# Beyond the Heart: The Predictive Role of Coronary Artery Calcium Scoring in Non-Cardiovascular Disease Risk Stratification

**DOI:** 10.3390/diagnostics14212349

**Published:** 2024-10-22

**Authors:** Viviana Cortiana, Hetvee Vaghela, Rahul Bakhle, Tony Santhosh, Oroshay Kaiwan, Aalia Tausif, Ashish Goel, Mohammed K. Suhail, Neil Patel, Omar Akram, Nirja Kaka, Yashendra Sethi, Arsalan Moinuddin

**Affiliations:** 1PearResearch, Dehradun 248001, India; viviana.cortiana@studio.unibo.it (V.C.); hetveev@gmail.com (H.V.); bakhlerahul577@gmail.com (R.B.); santhoshtony92@gmail.com (T.S.); aaliatausif@hotmail.com (A.T.); neil@pearresearch.com (N.P.); nirja@pearresearch.com (N.K.); drarsalan.moinuddin@gmail.com (A.M.); 2Department of Medical and Surgical Sciences (DIMEC), University of Bologna, 40126 Bologna, Italy; 3Pandit Deendayal Upadhyay Medical College, Rajkot 360001, India; 4Medical College Baroda, Maharaja Sayajirao University, Vadodara 390001, India; 5Dr. Somervell Memorial CSI Medical College, Thiruvananthapuram 695504, India; 6Department of Medicine, MetroHealth Medical Center, Case Western Reserve University, Cleveland, OH 44106, USA; 7Department of Physiology, Graphic Era Institute of Medical Sciences, Dehradun 248008, India; dr.ashishgoel2012@gmail.com; 8Department of Public Health & Community Medicine, International Medical University, Kuala Lumpur 57000, Malaysia; mohammedsuhail@imu.edu.my; 9Department of Medicine, GMERS Medical College, Himmatnagar 390021, India; 10Department of Medicine, Highland Hospital, Alameda Health System, Oakland, CA 94602, USA; 11Department of Medicine, Government Doon Medical College, HNB Uttarakhand Medical Education University, Dehradun 248001, India; 12School of Sports and Exercise, University of Gloucestershire, Cheltenham GL50 2RH, UK

**Keywords:** coronary artery calcification, coronary artery calcium score, calcium score, prognostic factors, cardiovascular disease

## Abstract

Coronary artery calcium scoring (CACS), a non-invasive measure of coronary atherosclerosis, has significantly enhanced cardiovascular (CV) risk assessment and stratification in asymptomatic individuals. More recently, a higher score for CAC has been associated with an increased risk of non-CV diseases and all-cause mortality. This review consolidated evidence supporting the role of CAC in assessing non-CV diseases, emphasizing its potential in early diagnosis and prognosis. We observed a strong association between CACS and non-CV diseases, viz., chronic obstructive pulmonary disease, pulmonary embolism, pneumonia, diabetes, chronic kidney disease, osteoporosis, metabolic dysfunction-associated steatotic liver disease, nephrolithiasis, stroke, dementia, malignancies, and several autoimmune diseases. Also, CAC may aid in evaluating the risk of CV conditions developing secondary to the non-CV diseases mentioned earlier. Further evidence from prospective studies, intervention trials, and population-based behavioral studies is needed to establish CAC cutoff values and explore preventative care applications, facilitating their broader integration into healthcare practices.

## 1. Introduction

Being a powerful risk predictor, coronary artery calcium (CAC) scoring is widely used as a reliable, non-invasive, and a replicable indicator for assessing the risk of atherosclerotic cardiovascular disease (ASCVD) [1]. Its application has significantly enhanced cardiovascular (CV) risk assessment and stratification in asymptomatic individuals [1,2,3,4]. Previous research correlates a yearly rise of ≥ 15% in coronary calcium volume with a 17-fold increase in CV event risk [5]. Furthermore, the risk of myocardial infarction (MI) was 17.2 times greater (95% CI: 4.1–71.2, *p* < 0.0001) with CAC progression as opposed to without it [5]. By combining traditional risk factors with CAC measurements, an accurate estimation of the 10-year risk of coronary heart disease (CHD) can be obtained as reported by a study that showed that the incorporation of CAC scores into the MESA (Multi-Ethnic Study of Atherosclerosis) significantly boosted the accuracy of CHD risk prediction (C-statistic 0.80 compared to 0.75; *p* < 0.0001) [3].

Notably, a CAC score of 0 corresponds to CV event rates below 5% over a 10-year period, while a CAC score greater than 100 corresponds to event rates above 7.5%. The specific CAC scores could be valuable thresholds for effectively communicating these cardiovascular risks to patients and facilitating discussions about risk-based treatments or preventative therapies [1,6,7]. In the past, numerous studies have extensively examined the connection between CAC scores and CV risk. However, recent evidence suggests that the implications of CAC extend beyond traditional cardiovascular conditions, offering potential as a predictive marker for a range of non-cardiovascular (non-CV) diseases. This extension of CAC utility is based on the broader concept that calcification is a systemic process often reflecting widespread arterial pathology, including conditions associated with chronic inflammation, metabolic dysregulation, and aging. But fewer studies focus on the possible correlation between CAC scoring and non-CV illnesses such as chronic kidney disease (CKD), chronic obstructive pulmonary disease (COPD), autoimmune diseases, stroke, dementia, and various malignancies like lung cancer with chronic inflammation and growth factors as possible underlying mechanisms. Indeed, compared to people with lower CAC values, those with very high CAC scores (≥1000) are likely to experience greater all-cause mortality. Despite this, the routine use of CAC in non-CV risk stratification remains limited. This limitation is primarily due to the lack of comprehensive guidelines and validation studies in these populations. Furthermore, the mechanisms linking CAC to non-CV diseases are still not fully understood. As such, assessing the risk of non-CV diseases using CAC scores is therefore of great importance to shed light on how the deposition of calcium affects systemic circulation to better target disease prevention interventions [8,9]. This review provides a perspective on (1) the link between CAC scoring and non-CV diseases, as well as (2) the potential of CAC scoring in evaluating the risk of CV conditions developing secondary to non-CV diseases.

## 2. Coronary Artery Calcification

### 2.1. Pathophysiology

Vascular calcification, in general, is a normal aging process that has been found to have strong pathological roots at a molecular level. Specifically, overexpression of transcription factors like RUNX2 or MSX2, commonly accounted for in atherosclerosis, can induce the de-differentiation of vascular smooth muscle cells (VSMCs) into osteoblasts or chondrocytes. These cells lay down collagenous and non-collagenous matrix at the site of vascular insult, which subsequently mineralize by the deposition of calcium and phosphorus, ultimately initiating the formation of a calcified plaque. The progression of CAC is shown in Figure 1. Furthermore, an imbalance between the inhibitors of calcification like Matrix GLA protein, osteocalcin, pyrophosphate, fetuin-A, and promoters of calcification like IL-6, IL-1, TNF-alpha, BMP-2, Cbfa1, high serum calcium, and phosphate levels are further responsible for the progression of microcalcification to densely calcified plaques [10,11,12,13].

### 2.2. CAC Quantification

The two distinct types of vascular calcifications are intimal and medial calcification. CAC, measured through a CT scan, however, cannot differentiate between them and hence calculates a cumulative calcium burden. The CAC score is typically quantified using three methods: the Agatston method, calcium volume score, and calcium mass score [14,15,16,17]. Whilst the Agatston method is the most widely utilized in clinical practice and research, the calcium volume score and calcium mass score have demonstrated greater consistency [18,19,20]. These methods are elaborated in Table 1.

### 2.3. Imaging Modalities for CAC Detection

Current CT scans can be completed within 10 to 15 min, exposing the patient to less than 1.5 millisievert (mSy) of radiation. The Society of Cardiovascular Computed Tomography recommends using this greatest effective dose for picture capture while computing CAC [17,23,24]. However, these methods have limitations, often overestimating calcified plaques due to blooming artifacts, particularly in heavily calcified cases. While it is useful for assessing calcified plaque burden, it cannot detect non-calcified plaques or provide information on plaque stability. Furthermore, CAC scoring involves radiation exposure, limiting its frequent use, especially in younger patients, and it lacks functional information regarding myocardial perfusion or ischemia. Ethnic and gender variations in CAC scores further complicate risk interpretation, and intermediate scores provide less clear guidance for clinical management. Advancements in geometric analysis and de-blooming algorithms are essential for enhancing coronary plaque evaluation in CT imaging [25]. Table 2 shows a comparison of different Imaging Modalities for CAC measurement [2,4,20,26,27,28].

### 2.4. CAC Scoring and CT Angiography

CT angiography (CTA) is not routinely recommended for asymptomatic individuals but plays a more valuable role in evaluating symptomatic patients or those undergoing coronary artery bypass grafting (CABG). On the other hand, CAC scoring can independently assess CAD risk in asymptomatic individuals and enhance CHD risk assessments. Table 3 is a comparison table between two such non-invasive imaging techniques, with CAC scoring using CT scans and CTA in terms of various aspects [6,26,29,30,31]:

### 2.5. Genetic Aspect of CAC in Non-Cardiac Diseases

Coronary artery calcification (CAC) is traditionally viewed as a cardiovascular risk marker, but its significance extends to non-cardiac diseases, driven by shared genetic predispositions. Key genes involved in lipid metabolism and vascular calcification, such as SORT1 and PCSK9, are linked to both coronary atherosclerosis and systemic conditions like chronic kidney disease (CKD) and osteoporosis [32,33]. In CKD, impaired mineral metabolism, particularly calcium and phosphate regulation, exacerbates vascular calcification, making CAC a predictor of cardiovascular and renal outcomes [34]. Furthermore, the genetic loci involved in inflammatory pathways, such as TNF and IL-6, also contribute to both atherosclerosis and chronic inflammatory diseases like rheumatoid arthritis, suggesting that CAC may reflect broader inflammatory and calcific processes in non-cardiac conditions [35]. Understanding the genetic underpinnings of CAC could help tailor therapeutic strategies for both cardiovascular and systemic diseases.

## 3. Endocrine Diseases

### 3.1. Diabetes Mellitus

Diabetes Mellitus (DM) is a metabolic disorder marked by elevated blood sugar levels brought on by perturbations in insulin production, function, or both. Prolonged hyperglycemia in DM can lead to organ damage and vascular calcification over time, with increased expression of osteogenic factors like ALP and BMP2 [36,37,38,39]. Additionally, systemic inflammation and hypercoagulability associated with DM are significant contributors to the development of CAD [40], accounting for approximately 70% of deaths in diabetics. As such, diabetics are at two to four times higher risk of adverse cardiac events [41] as reported by a 10.7-year follow-up study which found a strong association between CAC and CAD in type 1 diabetics (95% CI: 0.1–0.9, *p* < 0.027) [42]. Specifically, young diabetics face increased CAC risks due to factors such as male gender, smoking, and elevated Lp(a) levels [43], emphasizing the necessity of early CAD detection through CAC assessments [41]. Furthermore, as reported in the CARDIA study, women with a history of gestational DM (GDM) were twice more likely to develop CAC (score ≥ 100) within 15 years after pregnancy, compared to women with no history of gestational DM independent of their glucose status [44]. A family history of T2DM in these women further potentiates CVD risk at a younger age [45]. As such, screening methods like CAC are crucial for early detection and improved risk stratification of atherosclerotic CV events in women with a history of GDM, regardless of their glucose levels postpartum. Indeed, CAC emerges as a vital prognostic tool for diabetic patients with a coronary calcium score (CCS) ≥ 200 indicating unfavorable outcomes [46]. This finding is in contrast to a 14.7-year follow-up study (*n* = 9715) that suggested a CAC score of 0, indicating a favorable prognosis for 5 years in asymptomatic diabetics [47]. This was further confirmed by a prospective MESA study (*n* = 6814), which concluded that a CAC score of 0, regardless of the length of diabetes, use of insulin, or glycemic management, is linked to a lower risk of CVD [48], underscoring the role of CAC as an independent risk predictor [37]. Yet another multiethnic study investigating the relationship between retinopathy and an elevated CAC score in diabetics vs. non-diabetics [49] found a significant link between retinopathy and elevated CAC in DM patients but not in non-DM individuals (RR: 1.3, 95% CI: 1.02–1.66) suggesting that adding a CAC score to routine retinal exams in DM patients could be a simple but valuable screening tool for CVD in DM patients. The current American Diabetes Association preventive guidelines for CVD management in diabetics advocate aggressive treatment using statins in those with CAC scores ≥ 75th percentile [50,51]. Likewise, the American College of Cardiology (ACC) Cholesterol Treatment Guidelines recommend a class 2a treatment plan, i.e., CAC screening to assess CV risk in individuals with diabetes [42], even touting it as the most accurate non-invasive tool for risk stratification in asymptomatic DM patients [48]. It must be noted that CAC improves risk stratification and allows for appropriate preventive therapy based on CAC scores rather than the disease state. Refinement of risk stratification in diabetics with higher CAC can be achieved through functional stress testing [40].

### 3.2. Primary Hyperparathyroidism

Primary Hyperparathyroidism (PHPT) involves excessive parathyroid hormone production, which raises serum calcium and lowers serum phosphates. Despite this, symptomatic PHPT has been associated with arterial stiffness, hypertension, and CV risk; the CAC scores as such show no significant link with hyper/normo-calcemic hyperparathyroidism [52,53]. As noted by Kepez et al., mild PHPT alone does not contribute to coronary atherosclerosis; instead, traditional CV risk factors like hypertension and hyperlipidemia play a significant role [54]. To date, no large study has assessed CAC progression in PHPT patients, likely due to asymptomatic diagnoses and prompt treatment.

### 3.3. Sex Hormones

Intervention studies aimed at investigating the association of sex hormone levels (both testosterone and estrogen) with increased CV and all-cause mortality risk over a period of time yielded mixed results [55,56,57]. For example, Khazai et al. found low free testosterone levels associated with high CAC scores (>0), and low total testosterone levels with high log CAC scores [58]. Likewise, a multiethnic study in postmenopausal women found that those with initially no CAC but high free testosterone had an increased risk of incident CAC at follow-up (HR: 1.41, 95% CI: 1.03–1.92) and significant CAC progression (HR: 1.28, 95% CI: 1.03–1.59). The risk for CAC progression remained significant even after adjusting for CV risk factors while the risk for incident CAC became insignificant [59]. An imbalance between testosterone and estrogen (measured as T/E2 ratio) was also associated with inflammation and high IL-6 levels, both implicated in coronary artery calcification [60]. Taken together, it appears that serum testosterone levels influence CAC scores, and understanding this relationship can clarify the link between testosterone and CVD risk.

### 3.4. Aldosterone Levels

Despite the established premise of the association of elevated serum aldosterone with subclinical atherosclerosis, congestive heart failure (CHF) and ischemic heart disease (IHD), only a limited number of studies explore its relation to CAC. Inoue K and colleagues in their prospective study showed that high serum aldosterone levels (per 100 pg/mL) were associated with increased CAC scores, partly attributable to elevated blood pressure (RR: 1.17, 95% CI: 1.04–1.32), highlighting a possible independent association of aldosterone with vascular calcification [61]. Notably, primary aldosteronism increased the risk of non-CV disease like DM (OR: 1.33, 95% CI: 1.01–1.74) and metabolic syndrome (OR: 1.53, 1.22–1.91) compared to essential hypertension as concluded by another meta-analysis [62]. Thus, examining the link between rising aldosterone levels and CAC in larger studies could apparently yield valuable insights.

### 3.5. Osteoporosis

Atherosclerosis and osteoporosis share many common risk factors including aging, estrogen deficiency, smoking, low physical activity, diabetes, and many more. Evidence from clinical trials encompassing those have been instrumental in laying the foundation for plausible mechanistic links being connected by common pathways involving bone and vascular mineralization, inflammatory cytokines (IL-6 and TNF-α), and proteins like Matrix Gla and osteocalcin [63,64,65]. To cite this, a synthesized cross-sectional analysis of 186 postmenopausal women found that T-scores are independent predictors of coronary artery calcium scores (CACSs) (*p* < 0.05) [66]. Additionally, another study involving 5590 participants revealed that lower bone mineral density (BMD) levels and higher CACSs (CAC > 0) significantly correlated to an elevated risk of all-cause mortality (*p* = 0.0001) [67]. Similarly, another study from six MESA centers tracking 6814 participants for a median of 10.2 years found that higher CAC scores were associated with an elevated risk of hip fractures; proportions ranged from 0.26% (CAC = 0) to 1.77% (CAC > 400, *p* < 0.001). Indeed, doubling CAC increased hip fracture risk by 10% (1.01–1.21), while CAC > 400 raised risk (4.29, 1.47–12.5) and CAC = 0 reduced it by 69% (0.31, 0.14–0.70) [8]. In 863 postmenopausal women, high CACSs (≥100) and obstructive CAD were linked to osteoporosis, independent of CV risk factors and age [68]. To overcome the limitation of observational evidence ascribing causality, a juxtaposed meta-analysis (*n* = 10,299) found that low BMD independently predicts CAD (OR: 2.96, 95% CI: 2.25–3.88, *p* < 0.00001), whilst subclinical CV disease increases the risk of bone loss [65]. Nevertheless, another meta-analysis of 4170 participants from 11 studies showed contrasting results, suggesting that age might influence this association, as CAD patients had significantly lower BMD without age adjustment [69]. Chan et al. found an inverse correlation between trabecular BMD and coronary calcification in women but not in men; regardless, both sexes exhibited increased abdominal aortic calcification with decreasing BMD [70]. Wiegandt’s (Copenhagen General Population Study) findings suggest that lower BMD is positively correlated with a higher risk of a calibrated mass score greater than 0 in men and postmenopausal women, but not in premenopausal women, suggesting there could be a direct link between bone loss and atherosclerosis development, regardless of age and sex, although this might be proxy for deficient estrogen. Estrogen has been found to inhibit the calcification process through various mechanisms, including the modulation of inflammatory responses and lipid metabolism. However, postmenopausal women tend to exhibit higher CAC scores, most likely due to a decline in estrogen levels and the loss of the protective effect it confers. Furthermore, analogous to estrogen, testosterone in men has been associated with increased arterial calcification, contributing to an earlier onset of coronary artery disease. These sex-based hormonal influences provide important context when interpreting CAC scores in relation to osteoporosis and other diseases with known sex-specific prevalence and severity [71]. Future research should focus on stratifying these results as per age, sex, other population groups, and geographical locations from now on.

## 4. Neurological Diseases

### 4.1. Dementia

Dementia has a lengthy preclinical phase called mild cognitive impairment (MCI), bridging the onset of subtle cognitive decline and the diagnosis of dementia. Indeed, vascular injuries add up during this phase, negatively influencing the progression of dementia. Growing research shows that individuals with a high CACS had a higher likelihood of developing dementia. This risk heightens with increasing CACSs ranging from 0 to >1000 [72]. Findings reported to date suggest that dementia has the strongest association with lifetime stroke episodes, which can possibly be predicted using CACSs. Moreover, larger calcification volumes of the coronary, intracranial, and extracranial vessels have been proven to harm cognition and brain volumes as reported by part of the AGES-Reykjavik study that found worsened cognition and dementia to the degree of CAC accrual [73,74]. Since CAC represents the same atherosclerotic process causing calcification in cranial vessels, it is likely to be a competent indirect measure of cognitive dysfunction. Whilst the evidence is mixed, we would not contend that CACSs are precise (in totality) and unambiguously represent dementia; nevertheless, they offer a diligent insight into cognitive function. The prevalence of high dementia risk (CAIDE ≥ 10) increased with CAC: 4.67% (CACS = 0), 13.66% (CACS 1–399), and 24.79% (CACS ≥ 400, *p* < 0.001). CAC ≥ 400 was linked to higher dementia risk (OR = 2.30, *p* = 0.014), with moderate accuracy in identifying high-risk participants (AUC = 0.70, *p* < 0.001) [75]. Whilst a study of 532 individuals (mean age: 80 years) with a low or zero CACS had a significantly low long-term risk of incident dementia compared with those with CACS > 400 [76], another prospective study of the MESA cohort (*n* = 5570) reported no association between the progression of CACSs and dementia over a follow-up period of 2.5 years [77]. This result suggests that utilizing timely CACSs may not be beneficial for assessing individual responses to interventions. Nonetheless, the 2.5-year follow-up period is relatively limited and may not be clinically objective.

### 4.2. Cognitive Function and Psychosocial Well-Being

Emerging research suggests a potential link between CAC and cognitive as well as psychosocial health, highlighting the broader implications of cardiovascular disease beyond physical outcomes. A study examining 1332 dementia-free participants aged 40–80 from Beijing found that higher CAC scores were significantly linked to poorer cognitive outcomes, with correlations observed for both verbal memory (r = −0.083, *p* = 0.003) and global cognitive function (r = −0.070, *p* = 0.012) [75]. Similarly, in the SWAN (Study of Women’s Health Across the Nation) Heart Study, 312 women were assessed to explore whether psychosocial well-being influenced CAC progression. The results demonstrated that higher psychosocial well-being, evaluated through six validated questionnaires, was associated with reduced CAC progression (RR = 0.909, 95% CI, 0.843–0.979, *p* = 0.012), even after adjusting for potential confounders. These findings suggest a broader need for integrated prevention strategies that address both cognitive and mental health dimensions in relation to cardiovascular risk [78].

## 5. Pulmonary Diseases

### 5.1. Chronic Obstructive Pulmonary Disease

Chronic obstructive pulmonary disease (COPD) encompasses emphysema (damage to the air sacs in the lungs) and chronic bronchitis (long-term inflammation of the airways) that causes progressive expiratory airflow limitation [79]. Whilst the precise link between COPD and CAD remains unclear, growing evidence suggests that chronic low-grade inflammation may be involved [79]. Despite CAC and atherosclerosis being connected via raised inflammatory markers, a weak association of CAC with COPD implies distinct mechanisms [80]. At least two studies at Johns Hopkins University in the US have examined CACSs with risk of incident COPD. First, a multivariable-adjusted 10-year-long study by Handy et al. demonstrated a CACS > 400 was correlated with a 271% higher risk of COPD diagnosis. Also, a two-fold increase in a CACS heightened COPD risk by 10% [8]. Second, Peng A W et al. demonstrated that subjects with a CACS ≥ 1000 are associated with an increased risk of COPD diagnosis than those with a CACS of 400–999 [9]. Additionally, cross-sectional data from three studies, as logically expected, showed an inverse relationship between FEV_1_ (forced expiratory volume in one second) and CAC [81,82,83], particularly pertinent to smoking (a major COPD risk factor). The Evaluation of COPD Longitudinally to Identify Predictive Surrogate End-points (ECLIPSE) study of 942 subjects demonstrated a higher CACS in those with COPD compared to in smokers with normal spirometry and in non-smokers [84]. Furthermore, a 3-year follow-up of the ECLIPSE study revealed that subjects with higher COPD mortality had a higher CACS compared to that of COPD survivors [84]. The ECLIPSE study also suggests that a greater CACS in COPD is related to increased dyspnea, reduced exercise tolerance, and higher all-cause mortality [84]. In summary, whilst these are strongly indicative findings, there is a need to determine whether these associations are relevant.

### 5.2. Pulmonary Embolism

Pulmonary embolism (PE), defined as the sudden blockage of the pulmonary artery, is a significant cause of morbidity, particularly in hospitalized patients, and is a difficult pathology to diagnose clinically [85,86]. A multi-ethnic study by Handy et al. found no association between PE and the CACS, CAC > 400, or CAC = 0 [8], which agrees with another case–control study (with 200 subjects) which reported no significant CACS differences between patients with and without PE [87]. However, contrary to expectations, when these unrelated relationships were replicated prospectively in a 3-year follow-up study, the CACS cropped up as an incidental finding (on CT imaging) independently predicting all-cause mortality in patients with PE, alongside long-term mortality linked to the presence of co-existing CAD [88]. In addition, a separate multiple-variable-adjusted study demonstrated that individuals with CAC ≥ 1000 have an elevated risk of PE in comparison to those with CAC ≤ 400–999 or 0 [9]. It could be a confounding effect though, given that the majority of higher CACS findings in patients with PE coexist with CAD. Thus, further research involving patients with severe disease presentations is necessary to determine if an association exists in these study groups.

### 5.3. Pneumonia

Pneumonia, an acute respiratory illness caused by bacteria, viruses, fungi, and parasites, is the leading cause of infection-related death [89]. From the CAC–pneumonia association perspective, a multivariable-adjusted study investigating reported patients with CAC > 400 had a two-fold elevated risk of developing pneumonia, with a striking 7% increased risk for each doubling of the CACS compared to those with CAC = 0 [8]. Specific to COVID-19, atherosclerosis-related diseases such as hypertension, diabetes, and obesity are associated with increased COVID-19 severity [90,91]. To illustrate the overall pattern, a multitude of analyses suggest that the CACS assists in predicting COVID-19 prognosis. As expected, Takeshita et al. found CACS ≥ 180 to be associated with deteriorating oxygen levels and CT chest findings compared to other atherosclerosis risk factors like obesity, diabetes, and age in hospitalized patients (OR: 12.879, 95% CI: 1.399–380.401) [91]. Furthermore, a higher CACS in COVID-19 patients is linked to an increased likelihood of intubation and death compared to those without CAC [92]. Independent research supports these findings by showing a higher inpatient death rate among those with higher CACS [93]. Thus, CAC scoring may serve as a prognostic indicator for COVID-19 severity and the need for intubation.

### 5.4. Sarcoidosis

Sarcoidosis, a chronic inflammatory condition, can potentially lead to microvascular damage and early atherosclerosis, like other inflammatory diseases [94]. No known studies have investigated whether sarcoidosis leads to elevated CAC scores. However, a few acute studies have hinted mechanistic plausibility albeit demonstrating no significant difference in the CACS in patients with and without sarcoidosis [95]. Consequently, further investigations are required to validate such results and broaden the comprehension of this relationship.

## 6. Genitourinary Diseases

### 6.1. Chronic Kidney Disease

Chronic kidney disease (CKD) is a manifestation of structural and/or functional abnormalities of the kidneys causing a progressive decline in renal function. With a global prevalence of >10%, CKD is projected to be the fifth most common cause of years of life lost (YLL) by 2040 [96]. Most importantly, nearly 40–50% of deaths in patients with stage 4 CKD and end-stage renal disease (ESRD) are due to CVD or its associated complications [97,98]. CKD being proinflammatory causes intensive vascular and cardiac remodeling that can accelerate medial and intimal calcification, subsequently leading to an increased CAC load. Novel risk factors such as FGF-23, serum calcium, eGFR, 24 h urine albumin, IL-6, and TNF-α are associated with both deteriorating kidney function in CKD and CAC progression [99], as evident by the pooled prevalence of CAC in CKD patients to be 60% and higher in patients with ESRD undergoing dialysis [100]. Similarly, Budoff and colleagues demonstrated a significant graded association between the CAC score and lower kidney function with the odds ratio (OR) significantly increasing from 1.68 (95% CI, 1.23–2.31) to 2.82 (95% CI, 2.06–3.85) as the eGFR deteriorated from 50 to 59 mL/min/1.73 m^2^ to <30 mL/min/1.73 m^2^ [101]. A recent study from the United States found higher baseline CAC scores and progression to significantly amplify the serum calcification tendency [102]. This tendency has been associated with a heightened risk of all-cause mortality in CKD patients, regardless of their hemodialysis status [103,104]. Another prospective Chronic Renal Insufficiency cohort (CRIC) study (5.9 years of follow-up and involving 1541 non-dialysis CKD patients) reported that a 1 SD increase in the log CAC score significantly increased the risk of CVD, MI, and heart failure by 40%, 44%, and 39%, respectively, following adjustment for all novel cardiovascular and CKD-related risk factors. It is important to highlight that this study also found the inclusion of CAC scores to significantly predict CVD in CKD patients [105].

Multiple studies pointed out that there is no clear evidence of CAC reduction in hemodialysis patients and that dialysis alone does not reduce CVD risk in CKD patients, reiterating the need of additional interventions alongside hemodialysis [106]. Pharmacologically, non-calcium phosphate binders like sevelamer and the new drug SNF472 showed promise in slowing the progression of vascular calcification, yet their definitive impact on CKD prognosis requires more research [107]. As such, clinical trials to assess CAC as a CVD risk assessment tool in CKD and further investigation into the prognostic value of reducing vascular calcification in ESRD are needed.

### 6.2. Nephrolithiasis

The interest in understanding the association between nephrolithiasis and CAC has recently emerged following studies supporting the common risk factors (metabolic syndrome, DM, and CAD) they share [108,109,110]. Data collected from the MESA study showed that patients with recurrent kidney stones are at the highest risk of CAC [111]. Nevertheless, establishing clear evidence for the association between calcium and kidney stones, considering calcium as one of the major causes, remains a conundrum [112]. However, the significance of evidence on the elevated CAC score in patients (with a history of nephrolithiasis) is underscored by the findings of a recent study that reported individuals with a history of kidney stones have more extensive CAC, but no statistically significant association with coronary artery stenosis [113]. Furthermore, relative to patients without a history of kidney stones, patients with kidney stones had three times the risk of developing severe coronary calcification (CAC > 400). Overall, there is strong evidence of an association between kidney stone formation and subclinical coronary atherosclerosis, particularly in patients with high CAC levels [113].

### 6.3. Erectile Dysfunction

Research on erectile dysfunction needs to be extended beyond the research on endothelial dysfunction (measured by flow-mediated dilation, FMD), carotid intima-media thickness (cIMT), CAC, and additional vascular function measures, such as the ankle–brachial index, toe–brachial index, and pulse wave velocity [114], to understand the potential consequences of the CAC score pertained for CVD pathophysiology. Evidence supporting the association between erectile dysfunction (ED) and CVD was inconclusive, demanding that further exploration on this topic is imperative to address these outcomes. ED could serve as a simple yet effective subclinical screening indicator of CVD, particularly for younger men who are less likely to undergo CVD risk assessment.

## 7. Gastrointestinal Diseases

### Metabolic Dysfunction-Associated Steatotic Liver Disease (MASLD)

Metabolic dysfunction-associated steatotic liver disease (MASLD) is one the primary contributors to chronic liver disease, with an increasing global prevalence, most recently estimated to be 32% [115]. Although both CVD and MASLD have some overlapping risk factors like insulin resistance, obesity, and hyperlipidemia, the role of MASLD as an independent risk factor for CV disease is being increasingly investigated, and a strong consolidated association was proved in meta-analyses: first by Lu H et al. (OR: 1.50, *p* < 0.001), which prompted further investigation [116], and then by Jaruvongvanich V. et al., who compared 16,433 MASLD patients with 41,717 controls and demonstrated that patients with MASLD had 41% higher odds (OR: 1.41) of a CAC score > 0 and 24% higher odds (OR: 1.24) of a CAC score > 100 compared to controls [117]. This association was reaffirmed by Kapuria et al. (OR: 1.64), showing an increased occurrence of subclinical atherosclerosis in patients with MASLD [118]. Whilst the precise mechanism behind this association is unclear, several hypotheses attribute it in part to oxidative stress and insulin resistance (induced in MASLD) and the abnormal expression of liver biomarkers specific to atherosclerosis which together may increase the CV risk. More intriguingly, some genetic variants like PNPLA3 and TM6SF2 that increase MASLD risk have also been shown to mediate CVD risk. Other causes include an increase in carotid intima-media thickness [119] and the impaired endothelial function in MASLD [120]. Thus, including MASLD as an independent predictor in CVD risk calculators (e.g., ASCVD risk calculator) might envisage greater accuracy for CVD events; however, further studies need to solidify these hypotheses.

## 8. Psychological Disease

### 8.1. Schizophrenia

Studies examining the association between schizophrenia and CAC are limited; however, preliminary evidence reported CAC levels equal to zero among 163 schizophrenic patients, rendering CAC an unreliable indicator for CHD [6,121,122,123]. In contrast to this null finding, other studies found an increased risk of CHD in individuals with schizophrenia [124,125,126]. Such a discrepancy could be attributed to confounding variables such as smoking, which may not have been adequately accounted for in the studies that established a relationship between CHD and schizophrenia. Additional studies need to further explore this association.

The environmental correlates, such as lack of social support, adverse childhood experiences, early trauma, work-related stress, and social conflict [127,128,129,130,131,132], and psychological inclinations like anxiety, psychological stress, and depression (pertinent to the CACS and CVD mortality) reported perplexed results [133]. For instance, Smith et al. found connections between anxiety and anger with CAC, while depression showed no association, proposing the integration of structural models of negative affectivity and social behavior in enhancing the psychosocial risk factors associated with CAD. Furthermore, exclusive dependence on self-report measures for these traits may result in an underestimated CHD risk as demonstrated by another study investigating an association between concurrent asymptomatic CAD, evaluated through CAC, and hostility [134]. They enrolled 300 married couples (mean age: 54.4 years) without diagnosed CAD but found that hostility and antagonism were not related to CAC. Another study analyzing patients with different forms of hostility in relation to CAC (*n* = 571) reported only physical aggression as the parameter that could be related to CAC, i.e., a 5% increase in odds of CAC presence for every point increase in physical aggression [135]. These findings, indeed, have the potential to guide further investigations into the influence that environmental factors have on CVD, alongside personality traits to better understand the underpinning mechanisms involved and to validate the biological plausibility of CAC score perturbations being connected with psychiatric symptoms.

### 8.2. Anxiety and Depression

There is a well-established link between coronary artery disease (CAD) and mental health disorders, particularly depression and anxiety. A study in *Northern Clinics of Istanbul* found a significant correlation between increased CAC scores and elevated depression and anxiety levels in CAD patients [136]. This association may be driven by intersecting biological pathways, including chronic inflammation, autonomic dysfunction, and endothelial impairment, which contribute to both cardiovascular disease and psychiatric symptoms [137]. Depression has been shown to negatively affect cardiovascular outcomes, increasing mortality risk after myocardial infarction, which highlights the need for routine mental health evaluations in CAD patients [138]. Additionally, anxiety disorders are associated with higher sympathetic nervous system activity, which can exacerbate cardiovascular stress and worsen prognosis in CAD [139]. Addressing mental health in CAD patients through CAC score monitoring may therefore offer both psychological and cardiovascular benefits.

## 9. Cancers

Given that cancers account for nearly 10 million deaths each year worldwide, it is a priority for the CAC research agenda to garner novel evidence on the association between cancer and CAC scores. It has been known for a while that chemotherapy administered in cancer patients is correlated with an increase in atherosclerotic risk factors such as hypertension [140] and metabolic syndrome [141]. Additionally, radiotherapy has been shown to be associated with ischemic cardiovascular events [142,143,144]. This means there is a pressing need to build on these premises and explore a promising common pathway of chronic inflammation between the two variables [145], given that inflammation and coronary atherosclerosis are commonly observed together [146]. Moreover, it has been suggested that CAC could be helpful in predicting cancer incidence outcomes due to shared risk factors with CVD and because CAC represents a marker of tissue vulnerability (or resilience) to risk factor-related injury [147,148,149,150]. Addressing research on the correlation between cancer and CACS, Whitlock et al. investigated 3122 patients with no history of cancer or CVD [147]. In males, they found colorectal cancer to be the most frequently developed type (78%), and in those patients, CAC incidence was significantly higher than in the cancer-free comparison group. More importantly, CACSs have been identified as a risk factor, especially for specific types of cancer, with one of the strongest associations reported for colorectal cancer [151]. Thus, CAC might not only serve as a predictor for cardiac diseases but also to predict certain types of tumors. For example, in a recent trial involving 6271 patients with 777 cases of cancer, it was demonstrated that the cancer incidence rates per 1000 person-years were 13.1 for CAC = 0 and 35.8 for CAC ≥ 400. This bolstered the hypothesis of the predictive value of CAC for cancer incidence, possibly due to the risk factors shared with CVD. The simultaneous presence of atherosclerosis and colorectal adenoma has been hypothesized and further investigated [152]. Current results show a clear association between the two conditions: patients with colorectal adenoma had a higher odds ratio for high CAC, and vice versa. Specifically, the odds ratio for CAC > 100 in participants without colorectal adenoma was reported as 2.72, in low-risk colorectal adenoma 2.70, and in high-risk colorectal adenoma 3.11, compared to in patients without any colorectal adenoma. The results conclude a strong and independent association between CAC and colorectal adenoma. This finding could have implications for the screening of these conditions, suggesting that patients with CAC or colorectal adenoma should be checked for the other condition due to their increased risk.

### 9.1. Lung Cancer

A trial conducted by the Coronary Artery Calcium Consortium, involving a large cohort (*n* = 55,943), delved into the correlation between CAC levels and long-term mortality from lung cancer [153]. The rates per 1000 person-years were reported as 0.2 for CAC = 0 and 0.8 for CAC ≥ 400. Moreover, compared to in patients with no CAC, hazards for lung cancer mortality were higher in patients with CAC levels above 400. The strongest correlation was observed in smokers (both current and former) and women. These results not only provide additional insights into the association between CAC and lung cancer mortality but also help identify high-risk groups with elevated CAC levels. This knowledge can guide targeted preventive strategies and inform screenings. Currently, models for predicting lung cancer risk do not incorporate data from CAC scoring, making the findings from this trial particularly relevant and potentially impactful.

### 9.2. Nonfunctioning Adrenal Incidentalomas

Whilst there are still limited studies on nonfunctioning adrenal incidentalomas, the existing research results should encourage further exploration, which could yield interesting findings. Akkus et al. reported a significant association with CAC in 55 patients affected by nonfunctioning adrenal incidentaloma (NFAI) [154]. The NFAI group showed a higher CAC score than that of the control group, indicating a potentially higher risk of developing atherosclerosis. The study suggests that CAC levels in NFAI could be a relevant indicator for determining the risk of atherosclerosis.

## 10. Other Diseases

### 10.1. Autoimmune

Autoimmune diseases (AIDs), such as systemic lupus erythematosus (SLE), rheumatoid arthritis (RA), polymyositis, scleroderma, antiphospholipid syndrome, and others, are marked by persistent chronic inflammation, significantly accelerating atherosclerosis [155]. Consequently, such patients face up to a five-fold higher risk of CV events compared to the general population [156]. Studies that have examined the role of chronic inflammation, microvascular coronary dysfunction, and the cellular mechanisms underlying vascular calcification contributing to CAC pathogenesis provided compelling evidence [157]. For example, a meta-analysis of (*n* = 4500) patients [AM8] reported CAC tends to be higher in individuals with AIDs, possibly due to chronic exposure to proinflammatory molecules [158]. Furthermore, epidemiological evidence consistently shows an increased risk of CAD in patients with SLE [159,160,161]. In a recent meta-analysis of 24 studies on CAC prevalence in SLE patients, the random-effect prevalence was 29.8%, compared to 11.8% in controls (RR: 2.22, *p* = 0.0005) [162]. Additionally, another study revealed that SLE patients aged ≤ 45 years have a higher prevalence of detectable CAC compared to the general population [163]. Notably, the high prevalence of CAC > 0 in these patients is comparable to that seen in diabetic patients aged ≤ 40 years (32% vs. 43%, respectively) [164]. Nevertheless, there are wide gaps in the patterns of findings correlating SLE and CAC scores, with numerous questions remaining unanswered, thus suggesting the need for further investigation to explore potential parallels between these two conditions.

### 10.2. Psoriasis

Similarly to SLE, psoriasis too was found to be closely related to the CAC score as various preclinical studies propose a potential role of IL-17 and vascular inflammation [165,166]. It should be noted that patients with severe psoriasis face an elevated risk of major CV events [167,168]. In a recent cross-sectional analysis of 111 patients having severe chronic plaque psoriasis, approximately 86.2% were categorized as high/very-high-risk for CV events, with the remaining 25.6% not classified as high-risk. Upon assessing CAC scores amongst that 86.2%, 14 patients were reclassified as high/very-high-risk. When these cross-sectional findings were replicated via pooled evidence, involving 3039 psoriasis patients and 46,191 controls, a 54% increase in the CAC score was found between psoriasis and CAC (OR: 1.54, *p* < 0.00). This association was more pronounced in younger patients (mean age < 50, OR: 2.63, *p* < 0.001) than in older patients (OR: 1.24, *p* = 0.02). Indeed, further investigation is warranted to unravel the molecular pathways at play in the association between psoriasis and cardiovascular events [169].

### 10.3. Dental Pathologies

Dental conditions like caries, tooth loss, and periapical disease share risk factors such as low socioeconomic status, smoking, and diabetes with ischemic heart disease. Research indicates a higher prevalence of dental issues like caries and periodontal disease in individuals with acute MI compared to in matched groups [170]. Tooth loss due to periodontal disease is a risk factor for ASCVD, alongside factors like smoking and diabetes [171,172]. A retrospective cross-sectional study of 212 patients revealed a correlation between more severe dental issues and a higher CACS, but tooth loss was not an independent indicator of cardiovascular diseases [171,173]. Healthcare professionals should be aware that significant tooth loss might signal increased cardiovascular risk, necessitating further research into this connection and its underlying pathology.

Various key studies that report the use of CAC in non-cardiovascular diseases are listed in Appendix A. These studies emphasize the expanding relevance of CAC as a novel biomarker in diverse health conditions beyond traditional cardiovascular applications.

## 11. Challenges Limiting the Use of Coronary Artery Calcium Scoring in Non-CVD Patients

The use of CAC scoring in non-CVD patients is limited by several factors. First, current guidelines, such as those from the ACC/AHA, recommend its use primarily for intermediate-risk patients, but lack of consensus among major organizations restricts its application in broader non-CVD populations. Second, insurance coverage policy regulations such as CT Ca scoring are not often reimbursed for low-risk or asymptomatic individuals, further limiting its use [174]. Third, concerns related to radiation exposure and the potential for being over-diagnosed perhaps create hesitation among clinicians, who themselves may be unfamiliar with its utility for patients outside traditional cardiovascular risk groups [175]. Although research demonstrates that CT Ca scoring is a useful tool for stratifying risk in CVD patients, evidence supporting its effectiveness in non-CVD populations remains limited. Fourth, the absence of large-scale randomized controlled trials and lack of knowledge about recent developments have hindered its acceptability into routine practice. Fifth, the variability in scoring techniques, and potential overestimation due to calcification artifacts, and the challenges in interpreting CAC also remain significant challenges. Lastly, ethical concerns, including the potential for increased anxiety due to findings of subclinical calcification, also contribute to reluctance in its use. We advocate for addressing these challenges by requiring stronger evidence, particularly from robust clinical trials and expanded guidelines that clarify its role in preventive care beyond high-risk populations.

## 12. Limitations

While the association between CACSs and non-CVDs provides much-needed evidence, this is not without physiological and statistical limitations. First, evidence in this regard is still relative to some non-CVD pathologies. There is a possibility that studies with significant results are more likely to be published, while those with negative results may face publication bias. Further studies are needed to establish a consistent relationship between CACSs and non-cardiovascular diseases. Second, there is variability in CAC scoring across different studies. The use of various imaging techniques and scoring systems presents a challenge when attempting to compare results. Whilst most studies assessed CAC using ECG-gated CT and non-ECG-gated CT, studies on pulmonary embolism employed CT pulmonary angiography. Third, whilst quantitative Agatston scores were used in most studies, some employed volume-based scores. Fourth, the cutoff values of CACSs used in multiple studies differ. While a lower CAC score is typically considered indicative of low cardiovascular risk, higher scores are associated with increasing risk of cardiovascular events. However, for non-cardiovascular diseases, such as chronic kidney disease or cancer, the appropriate CAC cutoff remains unclear. Studies have used different thresholds to signify higher non-CVD risk, but these values vary depending on the patient population and the disease being studied. As CAC becomes increasingly used in risk stratification for non-CVD outcomes, standardizing these thresholds will be key to ensuring accurate and consistent application across clinical settings [9]. Standardizing imaging techniques and scoring systems would facilitate better comparison and improve risk prediction. Fifth, although many studies attempted to adjust for confounders, residual confounders may still exist, including age, gender, ethnicity, and comorbidities, viz., hypertension and hyperlipidemia, which can influence changes in the CACS and disease severity. Lastly, a relatively small sample size in some studies renders findings underpowered to establish a firm relationship.

## 13. Future Perspectives

This review unravels the untapped potential of CAC scores in predicting and monitoring health outcomes (worsening) in non-CVD pathologies. For instance, one possibility is to incorporate CAC scoring in patients with deteriorating kidney function in CKD, which is evident from this literature review. This should encourage the design of clinical trials with treatment options focusing on slowing down CAC progression and its impact on the prognosis of CKD patients, as hemodialysis, the current prognostic marker of CVD, alone carries no significant CVD-risk-reducing value. As no known study delved into stroke and COPD patients with a high CACS, we advocate that measuring the CACS in stroke and COPD might help it emerge as a subclinical risk marker. We saw that cancers like colorectal and lung cancers consistently show a high CAC incidence. We recommend the inclusion of the CACS in tools assessing the prognosis of these cancers. Furthermore, our study suggests a potential influence of testosterone, estrogen, and aldosterone levels on the CACS. This finding could serve as a basis for setting up clinical trials to establish a more definitive relationship between these hormones and CVD risk, which is currently a topic of debate in the medical community. Furthermore, there is much that remains unexplored concerning the potential usefulness of CACSs in post-operative patients or their application in assessing treatment progression. Together, surely there exists a wide array of opportunities for future research that could yield valuable insights into improving patient outcomes across a wide range of medical conditions.

## 14. Conclusions

CAC scoring, a non-invasive measure of coronary atherosclerosis derived from computed tomography scans, has significantly enhanced CV risk assessment and stratification in asymptomatic individuals. Recently, higher CACSs have been associated with an increased risk of non-CV diseases and all-cause mortality. This review consolidates evidence supporting the role of CAC in assessing non-CV diseases, emphasizing its potential in early diagnosis and prognosis. We observed a strong association between CACSs and several non-CV conditions. In COPD, elevated CACSs are linked to increased mortality and myocardial infarction. In PE and pneumonia, CACSs correlate with higher adverse outcomes, suggesting a broader systemic impact of underlying vascular calcification. Diabetes and CKD also show a significant association, with a higher CACS indicating greater risks for CV events and mortality. Moreover, osteoporosis has been connected to an increased CACS, possibly reflecting shared pathophysiological mechanisms involving calcification processes. Metabolic dysfunction-associated steatotic liver disease (MASLD) and nephrolithiasis are other conditions where the CACS has proven useful in risk stratification. Neurological conditions such as stroke and dementia also correlate with a higher CACS, underscoring the interconnectedness of vascular health and cognitive decline. Additionally, malignancies and several autoimmune diseases, including rheumatoid arthritis and systemic lupus erythematosus, show associations with an elevated CACS, highlighting the role of systemic inflammation in coronary calcification. Furthermore, the CACS may aid in evaluating the risk of CV conditions developing secondary to these non-CV diseases. Further evidence from prospective studies, intervention trials, and population-based studies is needed to establish CAC cutoff values and explore preventative care applications, facilitating their broader integration into healthcare practices.

## Figures and Tables

**Figure 1 diagnostics-14-02349-f001:**
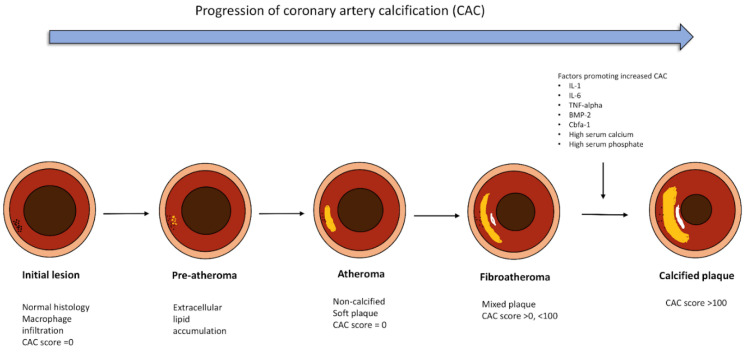
Progression of Coronary Artery Calcification: The figure shows how a coronary artery plaque progresses from an initial endothelial insult to formation of an atheroma and ultimately gets calcified to form a calcified plaque. Created using Biorender.

**Table 1 diagnostics-14-02349-t001:** Methods for CAC scoring.

Method	Description
Agatston Method	The CAC score totals calcium deposits above 130 Hounsfield units (HU), weighted by their highest density and ignoring plaque distribution [17,21].
Calcium Volume Score (CVS)	It measures actual CAC volume by multiplying voxel volume with >130 HU by the number of such voxels. It enhances reproducibility and reduces scan differences [22].
Relative Calcium Mass Score	It is the product of average calcified plaque density and plaque volume. It mitigates partial volume effects and reduces variation [22].

**Table 2 diagnostics-14-02349-t002:** Comparison of different Imaging Modalities for CAC measurement.

Imaging Method	Characteristics	Limitations
Initial Scanning Techniques	Utilized chest X-rays, fluoroscopy, digital subtraction fluoroscopy [27]	Potential for predicting CAD and coronary events, low sensitivity (52%) [14]
Electron-Beam CT (EBCT)	Introduced cardiac gating; higher sensitivity (90%) for CAC detection, localization, and quantification [14]; adequate temporal resolution for a moving heart [2]; quicker imaging through electronic movement [20]	Not suitable for general CT imaging, replaced by MDCT for CAC [2]
Multidetector CT (MDCT)	Allows reconstruction of a complete image [17], higher mean Hounsfield units (compared to EBCT) [28], superior spatial resolution, cost-effective option [20]	More significant motion artifacts (compared to EBCT) [28]

**Table 3 diagnostics-14-02349-t003:** CACS using CT scans and CTA.

Aspect	CAC Scoring	CT Angiography (CTA)
Imaging focus	Measures calcified coronary artery plaque	Visualizes calcified and noncalcified plaque, assesses luminal stenosis severity
Clinical indications	Risk assessment in asymptomatic individuals	Evaluation in symptomatic patients with suspected coronary heart disease (CHD)
Patient preparation	No IV access needed, requires breath-holding and remaining still	IV contrast administration, contraindication review, breath-holding, and regular heart rate preferred
Use of medications	Typically none	Beta blockers and nitrates for heart rate and artery dilation in certain cases
Impact on prognosis and clinical management	Improves CHD risk scores, helps reclassify at-risk asymptomatic individuals, and helps tailor preventive therapy (statins or aspirins) for high CAC scores	Identifies CAD spectrum, including non-obstructive CAD; requires further functional assessment for lesion significance and flow limitation

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
