# Peer review of "Beyond the Heart: The Predictive Role of Coronary Artery Calcium Scoring in Non-Cardiovascular Disease Risk Stratification"

_diagnostics, 2024, doi:10.3390/diagnostics14212349_

Round 1
Reviewer 1 Report
Comments and Suggestions for Authors
The diagnostic value of coronary calcification on non-cardiovascular diseases is an interesting topic. This review provides an overview on this topic. Overall, the manuscript is well structured and written. Some further improvements are essential to achieve publishable standards.
1. Some non-cardiovascular diseases have been identified as associated with coronary calcification but not surveyed, e.g., hip fracture (Refer: 10.1016/j.jcmg.2015.09.020). Please systematically search recent studies and enrich the content.
2. Regarding the evaluation of coronary artery calcification, the limitations of the state of the art need to be underscored. A major limitation is the blooming artifact which distorts the calcification regions and lead to the overestimation of calcification (Refer: 10.3389/fcvm.2021.597568).
3. Stroke is often deemed as a cardiovascular disease and is relevant to the whole-body atherosclerosis of which coronary calcification is an indicator. Therefore, I recommend to remove stroke and add some other neural and psychological diseases (e.g., 10.11909/j.issn.1671-5411.2021.07.002, 10.1161/JAHA.121.023937).
4. It is recommendable to add a table for stratifying the strength of evidence (e.g., meta analysis, clinical trial, retrospective study, animal/cell experiment, etc.) on different diseases.
Comments on the Quality of English Language
Overall the writing is OK but some sentences need further improvement, e.g., "..is widely 49 used as a, reliable, non-invasive, and a replicable indicator...". Further polishing is recommendable.
Author Response
We would like to extend our sincere gratitude for the valuable feedback provided by the reviewers. Their insightful comments have helped us improve the quality and clarity of our manuscript. We appreciate the time and effort you and the reviewers have invested in reviewing our submission.
In response to the reviewers' suggestions, we have revised the manuscript accordingly and addressed each of the points raised. Below, we provide a detailed response to each of their concerns.
Reviewer #1
- Some non-cardiovascular diseases have been identified as associated with coronary calcification but not surveyed, e.g., hip fracture (Refer: 10.1016/j.jcmg.2015.09.020). Please systematically search recent studies and enrich the content.
Authors' Response: Thank you for the suggestion. We have systematically searched recent studies to incorporate a sub-section in the discussion highlighting the association between coronary calcification and hip fractures. This includes references to the study mentioned (10.1016/j.jcmg.2015.09.020).
- Regarding the evaluation of coronary artery calcification, the limitations of the state of the art need to be underscored. A major limitation is the blooming artifact which distorts the calcification regions and leads to the overestimation of calcification (Refer: 10.3389/fcvm.2021.597568).
Authors' Response: We have added a brief snippet on the limitations of current methods evaluating coronary artery calcification, with particular attention given to the blooming artifact and its implications for overestimation (referencing 10.3389/fcvm.2021.597568).
- Stroke is often deemed a cardiovascular disease and is relevant to whole-body atherosclerosis, of which coronary calcification is an indicator. Therefore, I recommend removing stroke and adding other neural and psychological diseases (e.g., 10.11909/j.issn.1671-5411.2021.07.002, 10.1161/JAHA.121.023937).
Authors' Response: We agree with this recommendation. We have removed stroke from the non-cardiovascular diseases section and replaced it with a discussion on neural problems like cognitive decline and psychosocial wellbeing, including references to the studies provided (10.11909/j.issn.1671-5411.2021.07.002, 10.1161/JAHA.121.023937).
- It is recommendable to add a table for stratifying the strength of evidence (e.g., meta-analysis, clinical trial, retrospective study, animal/cell experiment, etc.) on different diseases.
Authors' Response: We have added a table that stratifies the strength of the evidence supporting the association between coronary calcification and various non-cardiovascular diseases as Supplementary Table 1. This table categorizes the evidence by study type (meta-analysis, clinical trials, retrospective studies, etc.) and adds other necessary details.
Reviewer 2 Report
Comments and Suggestions for Authors
CACS, a non-invasive measure of coronary atherosclerosis derived from computed tomography scans, has significantly improved CV risk assessment and stratification in asymptomatic individuals. This review brings together the evidence supporting the role of CAC in the assessment of non-CV disease and highlights its potential in early diagnosis and prognosis. The role of systemic inflammation in coronary calcification is emphasised in this study and the association with high CACS is demonstrated. The findings of the review are noteworthy and there are a few points to be noted:
1 - Please compose a paragraph on the genetic aspect of the effect of CAC in non-cardiac diseases.
2- For the expansion of the review, is there a role of CAC score in anxiety disorders and depression? For this purpose, write a paragraph on "The relationship between coronary artery disease and depression and anxiety scores. Northern clinics of Istanbul, 2020, 7.5." use and refer to this study
Author Response
We would like to extend our sincere gratitude for the valuable feedback provided by the reviewers. Their insightful comments have helped us improve the quality and clarity of our manuscript. We appreciate the time and effort you and the reviewers have invested in reviewing our submission.
In response to the reviewers' suggestions, we have revised the manuscript accordingly and addressed each of the points raised. Below, we provide a detailed response to each of their concerns.
Reviewer #2
- Please compose a paragraph on the genetic aspect of the effect of CAC in non-cardiac diseases.
Authors' Response: We have added a new paragraph discussing the genetic aspects of coronary artery calcification and its impact on non-cardiac diseases. Relevant studies on genetic predisposition and its relationship to CAC progression in non-cardiac conditions have been incorporated.
- For the expansion of the review, is there a role of CAC score in anxiety disorders and depression? For this purpose, write a paragraph on 'The relationship between coronary artery disease and depression and anxiety scores. Northern clinics of Istanbul, 2020, 7.5.' Use and refer to this study.
Authors' Response: We have included a new section addressing the relationship between coronary artery calcification and anxiety disorders and depression, incorporating findings from the study 'The relationship between coronary artery disease and depression and anxiety scores. Northern Clinics of Istanbul, 2020, 7.5,' as suggested.
Reviewer 3 Report
Comments and Suggestions for Authors
The manuscript covers the predictive role of coronary artery calcium scoring (CAC) in non-cardiovascular (CV) disease risk stratification. The role of CAC in CV risk prediction is well-established and captured in detail. Figures, tables, and abbreviations are clear, and the manuscript is well organized. However, the relevance of extending it to non-CV diseases needs more background in the introduction. Also, the current gap is very generic and does not provide the specific issue that prohibits the application of CAC for routine markers for non-CV diseases. CAC and its association with diseases are broadly clear except for the following points.
· In the osteoporosis section, it is hinted that men and women may have different correlations but not explained in detail. Also, it will be important to explain how sex hormones influence the development of CAC.
· Why is most of the work cited here cross-sectional? Doesn’t it limit the ability to infer relationships between CAC and non-CVD outcomes? Incorporating more longitudinal studies will strengthen the study.
· How do we address the different CAC cutoff scoring needs to be discussed in detail?
· What are the challenges currently clinicians are facing that limit their use for non-CVD patients in clinical practice (guidelines, policies, research, trials, etc.)? Having a separate paragraph will be valuable.
· What additional challenges may it introduce while considering non-CVD patients in terms of CAC quantification, CAC detection, and scoring?
Author Response
Reviewer #3:
- The manuscript covers the predictive role of coronary artery calcium scoring (CAC) in non-cardiovascular (CV) disease risk stratification. The role of CAC in CV risk prediction is well-established and captured in detail. Figures, tables, and abbreviations are clear, and the manuscript is well organized. However, the relevance of extending it to non-CV diseases needs more background in the introduction. Also, the current gap is very generic and does not provide the specific issue that prohibits the application of CAC for routine markers for non-CV diseases. CAC and its association with diseases are broadly clear except for the following points.
Author’s Response: As suggested, we have added a more detailed background in the introduction, emphasizing the relevance of extending CAC to non-CV diseases. Further, we have also elaborated and emphasised the specific challenges that hinder its widespread ramification and application as a routine marker for non-CV diseases, providing greater specificity to the gaps mentioned.
- In the osteoporosis section, it is hinted that men and women may have different correlations but not explained in detail. Also, it will be important to explain how sex hormones influence the development of CAC.
Author’s Response: We agree that our argument re osteoporosis has been constrained, so, as suggested we have expanded this section to delve deeper into the sex-based differences in correlations. Additionally, we have incorporated recent reports on the role of sex hormones in CAC development, elucidating the biological mechanisms that underlie these differences in CAC accumulation between men and women.
- Why is most of the work cited here cross-sectional? Doesn’t it limit the ability to infer relationships between CAC and non-CVD outcomes? Incorporating more longitudinal studies will strengthen the study.
Author’s Response: Thanks for pointing this. We appreciate your observation regarding the quality of the evidence (cross-sectional) cited. However, we feel that the use of CAC in non-CV diseases is a relatively novel realm in CV research and although it continues to expand rapidly, it’s still naïve in context of the longitudinal evidence. Still, as suggested, we have acknowledged the limitations of cross-sectional studies in establishing causal inferences and have incorporated higher-quality studies where available. Additionally, we have added a Supplementary Table 1, which compiles studies assessing the quality of evidence and outcomes, providing a clearer understanding of the predictive role of CAC in non-CVD conditions.
- How do we address the different CAC cutoff scoring needs to be discussed in detail?
Author’s Response: We have added a detailed statement in limitations section discussing the different CAC cutoffs used in studies and their implications for non-CVD patients. This section highlights how variable cutoffs might affect risk stratification and outcomes in non-CVD settings.
- What are the challenges currently clinicians are facing that limit their use for non-CVD patients in clinical practice (guidelines, policies, research, trials, etc.)? Having a separate paragraph will be valuable.
Author’s Response: We agree with this suggestion; hence, have included a separate paragraph (Section 11) discussing the pragmatic policy-related challenges that limit the adoption of CAC for non-CVD patients. Briefly, this section encompasses - the lack of comprehensive research, clinical guidelines, and reimbursement policies that hinder its wider clinical use.
- What additional challenges may it introduce while considering non-CVD patients in terms of CAC quantification, CAC detection, and scoring?
Author’s Response: Thank you for the detailed feedback. We have added details in the new section (Section 11) addressing the challenges specific to CAC quantification and scoring in non-CVD patients. This includes issues such as variability in scoring techniques, potential overestimation due to calcification artifacts, and the challenges in interpreting CAC scores in the context of non-CVD conditions.
We genuinely thank the reviewer for their insightful suggestions and believe that these have significantly enhanced the content and presentation of this manuscript; we look forward to your further evaluation.
Round 2
Reviewer 1 Report
Comments and Suggestions for Authors
Thanks for the update. My earlier comments have been well addressed. Please check the format and language during proofing.
Comments on the Quality of English LanguageOverall it is well written. Further polishing is needed in proofing stage.